# Toxigenic Genes, Pathogenic Potential and Antimicrobial Resistance of *Bacillus cereus* Group Isolated from Ice Cream and Characterized by Whole Genome Sequencing

**DOI:** 10.3390/foods11162480

**Published:** 2022-08-17

**Authors:** Rosa Fraccalvieri, Angelica Bianco, Laura Maria Difato, Loredana Capozzi, Laura Del Sambro, Domenico Simone, Roberta Catanzariti, Marta Caruso, Domenico Galante, Giovanni Normanno, Lucia Palazzo, Maria Tempesta, Antonio Parisi

**Affiliations:** 1Istituto Zooprofilattico Sperimentale della Puglia e della Basilicata (IZS PB), Via Manfredonia 20, 71121 Foggia, Italy; 2Experimental Zooprophylactic Institute of Apulia and Basilicata, 71121 Foggia, Italy; 3Department of Science of Agriculture, Food and the Environment (SAFE), University of Foggia, 71121 Foggia, Italy; 4Department of Veterinary Medicine, University Aldo Moro of Bari, Strada per Casamassima Km 3, 70010 Valenzano, Italy

**Keywords:** *Bacillus cereus* group, ice cream, cereulide toxin, heat-labile enterotoxins, whole genome sequencing (WGS), antimicrobial resistance (AMR)

## Abstract

*Bacillus cereus* is isolated from a variety of foods where it may cause food spoilage and/or food poisoning due to its toxigenic and pathogenic nature. In this study, we identified members of *B. cereus* groups in 65% of the ice cream samples analyzed, which were characterized based on multi locus variable number tandem repeats analysis (MLVA) and whole genome sequencing (WGS). The MLVA revealed that 36 strains showed different allelic profiles. Analyses of WGS data enabled the identification of three members of the *B. cereus* group: *B. cereus* sensu stricto, *B. mosaicus* and *B. thuringiensis*. Based on the multi locus sequence typing (MLST) scheme, the strains were classified in 27 sequence types (STs), including ST26 that causes food poisoning. Toxin genes’ detection revealed the presence of the genes encoding nonhemolytic enterotoxin (NHE), hemolysin BL (HBL), cytotoxin K (cytK) and cereulide (ces) in 100%, 44%, 42% and 8% of the strains, respectively. The identification of eleven antimicrobial resistance (AMR) genes predicted the resistance to five different antimicrobials, and the resistance to beta-lactam antibiotics was confirmed with a phenotypic antimicrobial test. Taken together, the results showed that the *B. cereus* strains isolated from ice cream were a potential hazard for consumer safety.

## 1. Introduction

The *Bacillus cereus* (*B. cereus*) group, named also as *B. cereus* sensu lato (*B. cereus* s.l.), is a gram-positive, rod-shaped, spore-forming [1] bacteria. The group includes bioterrorism agent *B. anthracis*, food-borne pathogen *B. cereus* sensu stricto (*B. cereus* s.s.), and biopesticide *B. thuringiensis*. The members belonging to *B. cereus* s.l. include genetically closed strains, and a recent study [2] proposed a novel taxonomic nomenclature for the *B. cereus* group useful to standardize the “genomospecies/subspecies/biovar” nomenclatural framework for more accurate species assignment.

The *B. cereus* group is widespread in the environment and it can contaminate feedstocks used for food preparation [3]. Their spores, which are able to survive dehydration and pasteurization processes, contaminate food through dehydrated and pasteurized ingredients [3]. Among contaminated foods, raw milk represents a hazard to food and consumer safety because spores can survive pasteurization treatments [4,5]. Other foods that pose the same concern are ready-to-eat products, such as ice cream, wherein *B. cereus* has been isolated as a pathogen [6].

The *B. cereus* group produces various types of toxins including insecticidal or pathogenic toxins, which can cause diseases such as diarrhea, vomiting or serious food poisoning and hemolysis, and even death [7,8,9,10]. There is known strong evidence of food-borne outbreaks sustained from *B. cereus* [3]. Interestingly, in 2019 the typical toxins produced by *B. cereus* were the agents frequently reported at the EU (European Union) level with a number of FBOs (food-borne outbreaks) that were twice as many as the FBO numbers due to toxins produced by other pathogens, including *C. perfringens* or *S. aureus*. Of concern is the association between *Bacillus cereus* human infection and deaths [11]. The contamination of foods with *B. cereus* s.l. may lead to food poisoning that usually manifests as emetic syndrome and/or diarrheal syndrome [12]. The former is caused by ingestion of the cereulide toxin, encoded by the *ces* genes [13,14,15]. The second is caused by three heat-labile enterotoxins: hemolysin BL (HBL), nonhemolytic enterotoxin (NHE) and cytotoxin K (cytK). Hemolysin BL is transcribed from the gene cluster *hbl*ABCD [16]. The nonhemolytic enterotoxin is encoded by gene cluster *nhe*ABC [17]. The cytotoxin K (cytK) protein, encoded by the genes *cyt*K-1/2, is highly cytotoxic. It is implicated as the primary toxin responsible for diarrhea [8,18]. In the diarrheal syndrome the vegetative cells of *B. cereus*, present in food, after colonizing the intestine begin to produce the enterotoxins [7].

The distinguishing markers between the species within the *B. cereus* group are encoded by genes contained in highly mobile genetic elements (plasmids) [5,19,20]. *Bacillus thuringiensis* carries the plasmid that includes insecticidal toxin genes (e.g., the *cry* gene encoding d-endotoxins, *Vip* genes encoding vegetative insecticidal protein or *Cyt* gene) [2], while *B. anthracis* harbors two large plasmids: pXO1, encoding the anthrax toxin complex and the virulence regulator AtxA; pXO2, encoding the virulence factor poly-g-D-glutamic acid capsule [21]. Moreover, the genetic determinants of the *B. cereus* emetic toxin, the gene cluster *ces*ABCD, are located on a large plasmid [5,22,23]. Although food-borne diseases associated with the consumption of *B. cereus*-contaminated foods do not necessarily require therapeutic treatment, investigating the genes involved in antibiotic resistance may be important. To date, the best strategy for performing typing and the genetic characterization of *B. cereus* s.l. is conferred by the WGS approach [3].

The aim of this study was to investigate the presence of *B. cereus* species and their genes encoding toxins in ice cream samples marketed in two southern Italian regions, Apulia and Basilicata. We have described the genetic characterization of the strains and the potential risk based on their genotypic diversity, pathogenic potential and AMR.

## 2. Materials and Methods

### 2.1. Isolation of B. cereus Group from Ice Cream Samples

A total of 51 ice cream samples were collected: 24 (24/51; 47%) homemade ice creams, collected from different local pastry shops and 27 (27/51; 53%) commercially produced (packaged) ice creams from 11 different Italian brands purchased from supermarkets in two southern Italian regions, Apulia and Basilicata. The samples were transferred to the laboratory in sterile bags. The detection of presumptive *B. cereus* was performed in accord to ISO 21871:2006 E [24]. Briefly, 5 g of food were homogenized in 45 mL Buffered Peptone Water (BPW) (BPW, Biolife, Milan, Italy) for 30 s; after, the sample was supplemented with 45 mL of Tryptic Soy Polimixin Broth (TSPB, Biolife, Milan, Italy), homogenized and heated at 30 °C for 48 h. An aliquot (10 μL) from each sample was plated directly onto Agar Mannitol Egg Yolk Polimixin (MYP, Biolife, Milan, Italy) and incubated at 30 °C for 24/48 h. From each plate, up to 5 presumptive strains possessing typical colony morphologies were confirmed as *B. cereus* using the hemolysis test on sheep blood agar plates. Bacterial strains isolated were stored at −80 °C.

### 2.2. DNA Extraction

Genomic DNA of each strain was purified using DNeasy Blood & Tissue (Qiagen, Hilden, Germany) following the manufacturer’s instructions. DNA concentration was determined by Qubit Quantitation using Qubit dsDNA HS Assay (Invitrogen, Thermo Fisher Scientific, Waltham, MA, USA).

### 2.3. Multi Locus VNTR Analysis (MLVA)

Five presumptive *B. cereus* strains, isolated from each positive sample, were screened by MLVA as previously reported by Valjevac et al., 2005 [25]. The products of polymerase chain reaction (PCR) were visualized by electrophoretic separation using QIAxcel High Resolution Kit on QIAxcel instruments (Qiagen). The different allele profiles identified in isolates derived from the same food source were selected for further characterizations by WGS.

### 2.4. Whole Genome Sequencing

For each selected *B. cereus* strain, paired-end genomic libraries and sequencing were processed as previously reported in Bianco et al., 2020 [26]. Draft genomes were submitted to BTyper3 tool [27], which performs in silico analysis to detect: identification of the closest strain, sequence type (ST) and clonal complex, phylogenetic group based on *panC* gene sequence and the major toxin factors. We used BTyper2 tool (Version 2.3.2; https://github.com/lmc297/BTyper, accessed on 9 July 2022) [28] for antimicrobial (AMR) gene detection.

### 2.5. Antimicrobial Susceptibility Test

Antimicrobial susceptibility of all *B. cereus* strains was evaluated using the Kirby–Bauer disk diffusion method according to the Clinical and Laboratory Standards Institute [29] for *S. aureus* [30]. Twelve antibiotics (Liofilchem^®^, Roseto degli Abbruzzi, Italy) were tested, including ampicillin (AMP, 10 μg), amoxicillin-clavulanic acid (AUG, 20 μg/10 μg), penicillin (P, 10 U), cephalothin (KF, 30 μg), imipenem (IPM, 10 μg), gentamicin (CN, 10 μg), kanamycin (K, 30 μg), erythromycin (E, 15 μg), vancomycin (VA, 30 μg), chloramphenicol (C, 30 μg), tetracycline (TE, 30 μg), clindamycin (DA, 2 μg). After incubating for 18 h at 36 ± 1 °C, the inhibition zones were measured and interpreted referring to the Clinical and Laboratory Standards Institute (CLSI, 2010) for *S. aureus* [30]. *Staphylococcus aureus* ATCC 29,213 and *E. coli* ATCC 25,922 were used as control strains.

### 2.6. Nucleotide Sequence Accession Numbers

The draft genomes of the *B. cereus* identified have been deposited in GenBank as BioProject PRJNA773646. The numbers of BioSample and accession ID are reported in Appendix A.

### 2.7. Statistical Analysis

Differences among the percentages of different isolates were compared using the Chi-square test (v2, *p* < 0.05) with Epi Info 3.3.2 software (Centers for Disease Control and Prevention, Atlanta, GA, USA).

## 3. Results

The *B. cereus* colonies showed the typical color morphology, being dull gray and opaque, with a rough matted surface and irregular perimeters. Based on bacterial growth, 65% (33/51) of the ice cream samples analyzed were *B. cereus* group positive, of which 18 (18/27; 67%) were packaged ice cream and 15 (15/24; 63%) were homemade ice creams (Table 1). No statistically significant difference was observed between the packaged ice cream group and the homemade ice cream group (*p* > 0.05). Among the 51 ice cream samples, only 4 (2 homemade and 2 packaged) were fruit dairy-free ice cream, and they were all *B. cereus* group negative; the other samples were prepared using milk as an ingredient. The results of the MLVA performed on five strains isolated from each ice cream (*n* = 33) that were positive for *B. cereus* s.l., showed 36 different allelic profiles. Specifically, the strains isolated from 30 ice cream samples showed a unique genetic profile for each source analyzed, whereas the strains isolated from three sources showed two different *B. cereus* s.l. profiles, respectively (Table 1 and Appendix A). We selected these 36 strains to perform genetic characterization by WGS. The draft genome sequence of the investigated *B. cereus* s.l. strains consisted of an average of 146 contigs comprising approximately 5.588.926bp. The average coverage was estimated at ∼92X. The overall G + C content of 36 isolates was 35% (Appendix A). Bioinformatic analysis performed by the BTyper3 tool confirmed that all strains belonged to the *B. cereus* group: 12 strains (33%) were identified as *B. cereus* s.s.; 12 (33%) strains were identified as *B. mosaicus*; 8 strains (22%) were identified as *B. mosaicus* subsp. *cereus*; 3 strains (8%) were identified as *B. mosaicus* subsp. *cereus* biovar *Emeticus*; 1 strain (3%) was identified as *B. cereus* s.s. biovar *Thuringiensis*. The MLST analysis identified 27 different STs (Table 1); among these we identified six new STs that were submitted on public database PubMLST (https://pubmlst.org/ accessed on 13 August 2022). Based on *panC* gene sequences, the strains were distributed among three different phylogenetic groups (Table 1): 17 strains (47%) were attributed to phylogenetic group III; 13 strains (36%) were assigned to phylogenetic group IV; 6 strains (17%) were assigned to phylogenetic group II. The degree of pathogenicity was assessed by searching for toxin- (Table 1) and antibiotic-resistant genes (Table 2). The BTyper3 tool identified the gene cluster *nhe* (nonhemolytic enterotoxin) and the *Sph* (sphingomyelinase) and *bpsE* genes (exo-polysaccharide capsule gene) in all strains analyzed; the gene *bpsH* (exo-polysaccharide capsule gene) was identified in 47% (17/36) of the strains; the gene cluster encoding diarrheal toxin Hbl was identified in 44% (16/36) of the strains; the gene *cytK2* was identified in 42% (15/36) of the strains; the gene cluster *cesABCD*, encoding the emetic toxin cereulide, was identified in 8% (3/36) of the strains; the *Vip4Aa1* gene, encoding vegetative insecticidal proteins, was identified in one strain (3%), which was identified as *B. thuringiensis*. Additionally, the antimicrobial resistance genes were predicted (Table 2). All strains harbored the genes conferring resistance to beta-lactam and fosfomycin antibiotics. Eighty-six percent (31/36) of the strains carried the *vanR-M* genes and 31% (11/36) carried the *vanZF-Pp*; other AMT genes identified with a frequency <25% were: *lsa (B)*, which encoded a protein that confers resistance to clindamycin; *vanS-Pt2*, *vanR-Pt* and *vanY-Pt2* genes, which encode proteins that confer vancomycin resistance; the *tetL* gene that confers tetracycline resistance. Additionally, phenotypic antimicrobial susceptibility testing was performed. The results of the antimicrobial tests are presented in Table 3. All *B. cereus* s.l. strains resulted in being sensitive to imipenem, gentamycin, kanamycin, vancomycin and chloramphenicol. Similarly, as predicted by the BTyper tool, the majority of the isolates were susceptible to erythromycin (89%), clindamycin (97%) and tetracycline (89%) with one exception; these results were in agreement with AMR gene prediction, which identified the *tetL* gene in the same strain (*B. cereus* s.s. BC561A). A similar result was obtained for beta-lactam resistance: all strains (100%) were resistant to ampicillin, 97% to penicillin and 94% to amoxicillin/clavulanic acid and cephalothin.

## 4. Discussion

Spores and vegetative cells of the *B. cereus* group are ubiquitous in the environment and, therefore, are commonly found in different kinds of unprocessed food [7]. Spores may survive the intense processing of dehydrated foods and the pasteurization temperature [4,5,6,31]. and they can also germinate, and reproduce at low temperatures due to the psychotropic feature [31,32]. These challenges make *B. cereus* a very important and frequently encountered bacterium in the food industry [31,33]. The presence of *B. cereus* s.l. and their toxins in foods poses a potential risk for food spoilage and to public health [31,32]. *B. cereus* s.l. are psychrotolerant microorganisms and their contamination of refrigerated stored foods raises a food safety problem [31,33]. Various studies reported different rates of *B. cereus* s.l. contamination: in dried food samples of different pulses and cereals 56% [34]; in pasteurized milk 27% to 47% [30,35,36,37]; in milk and dairy products 14% to 70% [38,39,40,41]; in ice cream 30% to 63% [6,31,38,41]. Our study investigated the presence of the *B. cereus* group in ice creams, which was estimated to be 65% (33/51); a similar prevalence (62.7%) was described in Messelhausser U. et al., 2010 [6]. With the aim of screening the different members of *B. cereus* s.l. isolated from the samples collected, we performed MLVA that identified 36 different *B. cereus* s.l. profiles. When we compared the genetic profiles with species identification based on WGS, the correlation was found for 33% (12/36) of strains (Appendix A). Probably, this result was due to the low number of samples analyzed. However, MLVA analysis allowed the selection of samples for next-generation sequencing. The genetic characterization performed by WGS identified three species among the *B. cereus* group: *B. cereus* s.s., *B. mosaicus* and *B. thuringiensis*. Interestingly, all *B. cereus* s.s. and the one *B. thuringiensis* identified belonged to phylogenetic group IV, whereas the *B. mosaicus* belonged to both phylogenetic group II and III. Additionally, we identified 27 STs, of which six have not been described previously. The most prevalent ST was ST142, which was identified in 11% (4/36) of the strains, followed by ST26 identified in 8% (3/36) of the strains. This result was in agreement with a previous study that analyzed milk samples [26]. Among the other STs, ST1084, ST126, ST92 and ST98 were identified in two strains, whereas the remaining STs were in one strain. Interestingly, ST26 is known as an ST that causes food poisoning with vomiting and includes clinically isolated strains [42,43], suggesting that the strains belonging to this ST are potentially damaging strains that may be present in ready-to-eat foods. The probable pathogenic power exhibited by the isolated strain was assessed by the detection of the virulence factors and AMR genes. The most common enterotoxin-encoding genes identified were *nhe* and sph according to previous reports [26,44]. Other major virulence factors were identified, as well as the gene cluster of *HBL* (*hbl*ABCD) encoded for the enterotoxin BL and *cyt*K2 (cytotoxin K) gene; these are genes that encode toxins with a high degree of cytopathogenicity [45,46,47]. Some previous studies reported that 35 to 100% of the strains isolated from various food items harbored *hbl* and *cyt*K genes [30,35,36,48,49,50,51,52,53]. The cytotoxic effect of enterotoxins and enzymes is determined by their combination and their levels of expression, strictly dependent on the strain [45]. Diarrheal syndrome is probably the result of the combination and synergistic action of different toxins and enzymes. The isolate identified as *B. thuringiensis* harbored the *Vip*4Aa1 gene that encoded the vegetative insecticidal proteins but did not harbor the *cry* gene that encoded δ-*cry* endotoxins responsible for the pesticidal action. The toxin responsible for the emetic syndrome is the heat stable cereulide protein encoded by the gene cluster *ces*ABCD [13], which we identified in 8% (3/36) of the analyzed isolates. Based on this finding, these strains can be considered potentially responsible for the emetic syndrome. Interestingly, these strains belonged to ST26, typically associated with emetic *B. cereus* isolates [54]. The finding of the prevalence rate of the gene *cesABCD* in *B. cereus* s.l. agreed with previous studies that isolated *B. cereus* s.l. from milk, dairy products and other sources [35,48,53]. The antibiotic susceptibility of bacteria is a public health concern. We identified genes that were potentially associated with the resistance of five different antibiotics, and we compared this result with the phenotypic test. The Fcyn-fosBx1 gene was identified in all isolates, which could confer Fosfomycin resistance; because there are no available references of phenotypic interpretation about the Fosfomycin resistance, this finding cannot be compared. Thirty-four isolates harbored the genes associated with vancomycin (*van*S-Pt2, *van*R-M, *van*R-Pt, *Van*ZF-Pp, *Van*R-F and *van*Y-Pt2) resistance, but all isolates resulted in being susceptible to the phenotypic test. Nine isolates carried the clindamycin (*lsa*B)-resistant gene, but all isolates were susceptible (except one that presented an intermediate resistance) to the phenotypic test. In one isolate we identified a gene *tet*L that confers resistance to tetracycline, and this finding was in accordance with the phenotypic test. In all isolates, we identified the genes *BLA*-1 and *BLA*-2 associated with the beta-lactam antibiotic; additionally, we found that all isolates were resistant to ampicillin, 94% resistant to amoxicillin-clavulanic acid, 97% resistant to penicillin G and 94% resistant to cephalothin using the phenotypic test. Typically, *B. cereus* is resistant to penicillin G or other beta-lactam antibiotics [55,56,57]. The ability to resist beta-lactam antibiotics may be conferred by the capability of *B. cereus* s.l. to synthesize the enzyme involved in the antibiotic degradation [57,58]. In *B. cereus* s.l., the production of this enzyme can lead to resistance even up to the third generation of cephalosporins [58,59]. Similarly to the results of other studies [30,57,58,60], the isolates tested showed susceptibility to imipenem, gentamicin, kanamycin, erythromycin, chloramphenicol, tetracycline and clindamycin. However, no gene associated with these resistances has been identified. These results highlighted that there is not always correlation between phenotype and genotype; in this study we evaluated the genetic profile of resistance genes, but we did not have any information about the genetic expression. Conversely, not all genes associated with AMR are known; thus, our study was in agreement with the concept reported in [45], which stressed the importance of updating dedicated databases with information derived from comparing AMR genes identified by WGS and antibiotic resistance detected by phenotypic testing.

## 5. Conclusions

Our study showed that the *B. cereus* group strains, isolated from ice cream, were potential enterotoxin producers and were also capable of exhibiting antibiotic resistance. *B. cereus* s.l. is a potential risk for the consumer due to its toxin-forming ability.

Interestingly, we identified three members of the *B. cereus* group, and the analyses revealed that *B. thuringiensis* and *B. mosaicus* can carry similar virulence determinants as *B. cereus* s.s. This study confirmed the low prevalence rate of emetic toxin gene *ces* and a high presence of the three heat-labile enterotoxins: hemolysin BL (HBL), nonhemolytic enterotoxin (NHE) and cytotoxin K (cytK). Furthermore, our results highlighted the resistance to penicillin and third-generation cephalosporins and sensitivity to other antibiotics tested.

We suggest that MLVA can be used for the purpose of carrying out a preliminary screening, and the analysis of WGS, followed by appropriate data analysis strategies, could be a highly effective way to evaluate the pathogenic potential of the *B. cereus* group.

## Figures and Tables

**Table 1 foods-11-02480-t001:** **Phylogenetic characterization of the 36 isolates of *B. cereus* s.l.** The table shows the data relating to the species identification, phylogenetic cluster, sequence type (ST), clonal complex and major toxin-virulence genes. In grey are indicated the strains isolated from three ice creams that each showed two different *B. cereus* s.l. profiles. In bold are indicated the new STs identified in the current study. (+): presence; (-): absence.

Source	Sample N. of Positive/N. of Total; Prevalence %	ID Strains	Taxon Names	*panC* Group	PubMLST ST	Clonal Complex	*Virulence Factors*							
							*ces*ABCD	*nhe*ABC	*hbl*ABCD	*CytK*	*Sph*	*BpsE*	*BpsH*	*Vip*4Aa1
**Packaged ice cream**	**18/27; 67%**	BC56B	*B. cereus* s.s.	Group_IV	197	ST-23	-	+	+	+	+	+	-	-
		BC147A	*B. mosaicus* subsp*. cereus; B. cereus*	Group_III	**2663**	ST-205	-	+	-	+	+	+	+	-
		BC384A	*B. mosaicus* subsp*. cereus biovar Emeticus*	Group_III	26	nd	+	+	-	-	+	+	-	-
		BC391A	*B. cereus* s.s.	Group_IV	142	ST-142	-	+	+	+	+	+	-	-
		BC429A	*B. mosaicus*	Group_III	127	nd	-	+	-	-	+	+	+	-
		BC430C	*B. cereus* s.s.	Group_IV	857	nd	-	+	+	+	+	+	+	-
		BC430D	*B. cereus* s.s.	Group_IV	98	nd	-	+	+	+	+	+	+	-
		BC431A	*B. cereus* s.s.	Group_IV	98	nd	-	+	+	+	+	+	+	-
		BC431B	*B. mosaicus*	Group_III	92	nd	-	+	-	-	+	+	+	-
		BC432C	*B. mosaicus* subsp*. cereus* biovar *Emeticus*	Group_III	26	nd	+	+	-	-	+	+	+	-
		BC433A	*B. mosaicus*	Group_III	**2681**	nd	-	+	-	-	+	+	+	-
		BC433C	*B. cereus* s.s.	Group_IV	1282	ST-142	-	+	-	+	+	+	-	-
		BC434A	*B. mosaicus*	Group_II	**2682**	nd	-	+	-	-	+	+	+	-
		BC435A	*B. cereus* s.s.	Group_IV	**2683**	nd	-	+	+	+	+	+	+	-
		BC477A	*B. mosaicus* subsp*. cereus; B. cereus*	Group_III	1084	nd	-	+	-	-	+	+	-	-
		BC478A	*B. mosaicus* subsp*. cereus; B. cereus*	Group_III	144	nd	-	+	-	-	+	+	-	-
		BC479A	*B. mosaicus* subsp*. cereus; B. cereus*	Group_III	1989	ST-205	-	+	-	+	+	+	-	-
		BC480A	*B. mosaicus*	Group_III	120	nd	-	+	-	-	+	+	-	-
		BC550A	*B. mosaicus* subsp*. cereus; B. cereus*	Group_III	205	ST-205	-	+	-	-	+	+	-	-
		BC559A	*B. mosaicus*	Group_II	**2727**	nd	-	+	-	-	+	+	+	-
		BC571A	*B. mosaicus*	Group_II	216	nd	-	+	+	-	+	+	+	-
**Homemade ice cream**	**15/24; 63%**	BC194C	*B. cereus* s.s.	Group_IV	142	ST-142	-	+	+	+	+	+	-	-
		BC437A	*B. cereus* s.s.	Group_IV	142	ST-142	-	+	+	+	+	+	-	-
		BC514A	*B. mosaicus* subsp*. cereus; B. cereus*	Group_III	1084	ST-205	-	+	-	-	+	+	-	-
		BC551A	*B. mosaicus*	Group_II	126	nd	-	+	+	-	+	+	+	-
		BC552A	*B. mosaicus* subsp*. cereus* biovar *Emeticus*	Group_III	26	nd	+	+	-	-	+	+	-	-
		BC553A	*B. mosaicus*	Group_II	126	nd	-	+	+	-	+	+	+	-
		BC557A	*B. mosaicus* subsp*. cereus; B. cereus*	Group_III	**2726**	nd	-	+	-	-	+	+	-	-
		BC558A	*B. cereus* s.s.	Group_IV	1967	nd	-	+	+	+	+	+	+	-
		BC560A	*B. mosaicus* subsp*. cereus; B. cereus*	Group_III	164	nd	-	+	-	-	+	+	-	-
		BC561A	*B. cereus* s.s.	Group_IV	33	nd	-	+	+	+	+	+	+	-
		BC563A	*B. cereus* s.s.	Group_IV	142	ST-142	-	+	+	+	+	+	-	-
		BC565A	*B. cereus* s.s. biovar *Thuringiensis*	Group_IV	138	ST-18	-	+	+	+	+	+	-	+
		BC567A	*B. mosaicus*	Group_III	92	nd	-	+	-	-	+	+	+	-
		BC568A	*B. mosaicus*	Group_III	365	ST-365	-	+	+	-	+	+	-	-
		BC569A	*B. mosaicus*	Group_II	369	nd	-	+	-	-	+	+	-	-

**Table 2 foods-11-02480-t002:** Occurrence of antimicrobial resistance genes identified in the strains of *B. cereus* s.l. In the table are the AMR genes predicted by the BTyper tool. (+): presence; (-): absence.

Source	Strains	ID Strains	*fosBx1*	*BLA2*	*BLA-1*	*lsaB*	*vanS-Pt2*	*vanR-M*	*vanR-Pt*	*VanZF-Pp*	*VanR-F*	*vanY-Pt2*	*tetL*
**Packaged ice cream**	*B. cereus* s.s.	BC56B	+	+	+	+	-	+	-	-	-	-	-
	*B. mosaicus* subsp*. cereus; B. cereus*	BC147A	+	+	+	-	-	+	-	-	-	-	-
	*B. mosaicus* subsp*. cereus* biovar *Emeticus*	BC384A	+	+	+	-	-	+	-	-	-	-	-
	*B. cereus* s.s.	BC391A	+	+	+	-	-	+	-	+	-	-	-
	*B. mosaicus*	BC429A	+	+	+	-	-	+	-	-	-	-	-
	*B. cereus* s.s.	BC430C	+	+	+	+	+	+	+	-	-	-	-
	*B. cereus* s.s.	BC430D	+	+	+	-	-	+	-	-	-	-	-
	*B. cereus* s.s.	BC431A	+	+	+	+	-	+	-	-	-	-	-
	*B. mosaicus*	BC431B	+	+	+	-	-	-	-	+	-	-	-
	*B. mosaicus* subsp*. cereus* biovar *Emeticus*	BC432C	+	+	+	-	-	+	-	-	-	-	-
	*B. mosaicus*	BC433A	+	+	+	-	-	-	-	+	-	-	-
	*B. cereus* s.s.	BC433C	+	+	+	-	-	+	-	+	-	-	-
	*B. mosaicus*	BC434A	+	+	+	-	-	+	-	-	-	-	-
	*B. cereus* s.s.	BC435A	+	+	+	-	-	+	-	-	-	-	-
	*B. mosaicus* subsp*. cereus; B. cereus*	BC477A	+	+	+	-	-	+	-	-	-	-	-
	*B. mosaicus* subsp*. cereus; B. cereus*	BC478A	+	+	+	-	-	+	-	-	-	-	-
	*B. mosaicus* subsp*. cereus; B. cereus*	BC479A	+	+	+	-	-	+	-	-	-	-	-
	*B. mosaicus*	BC480A	+	+	+	-	-	-	-	-	-	-	-
	*B. mosaicus* subsp*. cereus; B. cereus*	BC550A	+	+	+	-	-	+	-	+	-	-	-
	*B. mosaicus*	BC559A	+	+	+	-	-	-	-	-	-	-	-
	*B. mosaicus*	BC571A	+	+	+	+	-	+	-	-	-	-	-
**Homemade ice cream**	*B. cereus* s.s.	BC194C	+	+	+	-	-	+	-	+	-	-	-
	*B. cereus* s.s.	BC437A	+	+	+	-	-	+	-	+	-	-	-
	*B. mosaicus subsp. cereus; B. cereus*	BC514A	+	+	+	-	-	+	-	-	-	-	-
	*B. mosaicus*	BC551A	+	+	+	+	+	+	+	-	-	-	-
	*B. mosaicus* subsp*. cereus biovar Emeticus*	BC552A	+	+	+	-	-	+	-	-	-	-	-
	*B. mosaicus*	BC553A	+	+	+	+	+	+	+	-	-	-	-
	*B. mosaicus* subsp*. cereus; B. cereus*	BC557A	+	+	+	-	-	+	-	-	-	-	-
	*B. cereus* s.s.	BC558A	+	+	+	-	-	+	-	+	-	-	-
	*B. mosaicus* subsp*. cereus; B. cereus*	BC560A	+	+	+	-	-	+	-	-	-	-	-
	*B. cereus* s.s.	BC561A	+	+	+	+	+	+	+	-	-	+	+
	*B. cereus* s.s.	BC563A	+	+	+	-	-	+	-	+	-	-	-
	*B. cereus* s.s. biovar *Thuringiensis*	BC565A	+	+	+	+	-	+	-	-	-	-	-
	*B. mosaicus*	BC567A	+	+	+	-	-	-	-	+	-	-	-
	*B. mosaicus*	BC568A	+	+	+	+	+	+	+	+	-	-	-
	*B. mosaicus*	BC569A	+	+	+	-	-	+	-	-	-	-	-

**Table 3 foods-11-02480-t003:** Antimicrobial resistance pattern of the B. cereus s.l. strains.

	Beta-lactamase				Carbapenems	Aminoglycosides		Macrolides	Glycopeptides	Amphenicols	Tetracyclines	Lincosamides
ID Strains	AMP	AUG	P	KF	IMI	CN	K	E	VA	C	TE	DA
BC56B	R	R	R	R	S	S	S	I	S	S	I	S
BC147A	R	R	R	R	S	S	S	I	S	S	S	S
BC384A	R	R	R	S	S	S	S	S	S	S	S	S
BC391A	R	R	R	R	S	S	S	S	S	S	S	S
BC429A	R	R	R	R	S	S	S	S	S	S	S	S
BC430C	R	R	R	R	S	S	S	S	S	S	S	S
BC430D	R	R	R	R	S	S	S	S	S	S	S	S
BC431A	R	R	R	R	S	S	S	S	S	S	I	S
BC431B	R	R	R	R	S	S	S	S	S	S	S	S
BC432C	R	R	R	R	S	S	S	S	S	S	S	S
BC433A	R	R	R	R	S	S	S	S	S	S	S	S
BC433C	R	R	R	R	S	S	S	S	S	S	I	S
BC434A	R	R	R	R	S	S	S	S	S	S	S	S
BC435A	R	R	R	R	S	S	S	S	S	S	S	S
BC477A	R	R	R	R	S	S	S	I	S	S	S	S
BC478A	R	R	R	R	S	S	S	S	S	S	S	S
BC479A	R	R	R	R	S	S	S	S	S	S	S	S
BC480A	R	R	R	R	S	S	S	S	S	S	S	S
BC550A	R	R	R	R	S	S	S	S	S	S	S	S
BC559A	R	R	R	R	S	S	S	S	S	S	S	S
BC571A	R	R	R	R	S	S	S	S	S	S	S	S
BC194C	R	R	R	R	S	S	S	S	S	S	S	S
BC437A	R	R	R	R	S	S	S	S	S	S	S	S
BC514A	R	S	R	R	S	S	S	S	S	S	S	S
BC551A	R	R	R	R	S	S	S	S	S	S	S	S
BC552A	R	R	R	R	S	S	S	S	S	S	S	S
BC553A	R	S	S	S	S	S	S	S	S	S	S	I
BC557A	R	R	R	R	S	S	S	S	S	S	S	S
BC558A	R	R	R	R	S	S	S	S	S	S	S	S
BC560A	R	R	R	R	S	S	S	S	S	S	S	S
BC561A	R	R	R	R	S	S	S	S	S	S	R	S
BC563A	R	R	R	R	S	S	S	S	S	S	S	S
BC565A	R	R	R	R	S	S	S	S	S	S	S	S
BC567A	R	R	R	R	S	S	S	S	S	S	S	S
BC568A	R	R	R	R	S	S	S	I	S	S	S	S
BC569A	R	R	R	R	S	S	S	S	S	S	S	S

AMP: ampicillin; AUG: amoxicillin/ac.clav; P: penicillin; KF: cephalothin; IMI: imipenem; CN: gentamicin; K: kanamycin; E: erythromycin; VA: vancomycin; C: chloramphenicol; TE: tetracycline; DA: clindamycin.

## Data Availability

The data that support the findings of this study are available from the corresponding author upon reasonable request.

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
