# Peer review of "Toxigenic Genes, Pathogenic Potential and Antimicrobial Resistance of Bacillus cereus Group Isolated from Ice Cream and Characterized by Whole Genome Sequencing"

_foods, 2022, doi:10.3390/foods11162480_

Round 1

Reviewer 1 Report

In this manuscript the authors tested 51 ice cream samples for Bacillus cereus strains using multiple-locus variable number of tandem repeat analysis and whole genome sequencing.  The samples were 24 homemade ice creams from different local pastry shops and 27 commercially produced ice creams from different supermarkets in two southern regions of Italy.  The isolated strains were potential producers of enterotoxins and exhibited antibiotic resistance characteristics, which are potential risks for the consumer.

Whilst the sample size used in this study is small, and ideally should have been at least 300,  the text and tables are subjected to many corrections and changes, as follows:

Line 3.  Italicise “Bacillus cereus”

Line 17.  Remove “the” before “65%”

Line 18.  Define “MLVA” and insert “(WGS)” after “sequencing”

Line 20.  Define “MLST”

Line 21.  Define “STs” and change “cause” to “causes”

Line 22.  Change “reveal” to “reveals”

Line 24.  Define “AMR”

Line 27.  Change “exhibiting” to “exhibited” and “then they” to “which are”

Line 38.  Change “Bacillus cereus” to “B. cereus”

Line 41.  Insert “The” before “B. cereus”

Line 42.  Remove “the” before “food”

Line 48.  Insert “the” before “B. cereus”

Line 49.  Insert “the” before “B. cereus”

Line 52.  Change “on” to “in” and “toxin” to “toxins”

Line 55.  Change “pathogens as well as C. perfringens or S. aureus” to “pathogens, including C. perfringens and S. aureus”

Line 56.  Change “Bacillus cereus” to “B. cereus”

Line 58.  Change “the emetic, and/or the diarrheal syndrome” to “emetic syndrome and/or diarrheal syndrome”

Line 59.  Remove “the” before “ingestion”

Line 66.  Change “of” to “for”

Line 78.  Change “consist in the sequencing of entire” to “involves sequencing of the entire”

Line 79.  Change “the study” to “this study”

Line 81.  Insert “have” before “described”

Line 88.  Change “supermarket” to “supermarkets”

Line 89.  Change “taken” to “transferred”

Lines 90-93.  Provide a reference

Line 97.  Change “were” to “was”

Line 98.  Change “5” to “five”

Line 99.  Insert “the” before “hemolysis”

Line 105.  Change “Locus VNTR Analysis” to “locus VNTR analysis”

Line 106.  Italicise “B. cereus”

Line 107.  Change “Multiple-Locus Variable” to “multiple-locus variable” and “Analysis” to “analysis”

Line 108.  Change “Polymerase Chain Reaction” to “polymerase chain reaction”

Line 112.  Change “Genome Sequencing” to “genome sequencing”

Line 114.  Change “previous” to “previously” and insert reference number for Bianco et al

Line 115.  Change “closer” to “closest”

Line 128.  Change “Staphylococcus aureus” to “S. aureus”

Line 129.  Change “Staphylococcus aureus” to “S. aureus”

Line 130.  Change “Sequence Accession Numbers” to “sequence accession numbers”

Line 131.  Insert “the” before “B. cereus”

Line 139.  Italicise “B. cereus”

Line 142.  Change “creams” to “cream”

Line 143.  Italicise “B. cereus”

Line 145.  Italicise “B. cereus”

Line 147.  Change “sources” to “source”

Line 148.  Change “3” to “three” and italicise “B. cereus”

Line 149.  Change “performer” to “perform”

Line 150.  Change “whole genome sequencing” to “WGS”

Lines 150-151.  Italicise “B. cereus”

Line 153.  Insert “the” before “BTyper3”

Line 154.  Insert “the” before “B. cereus” and italicise “B. cereus”

Line 155.  Italicise “B. cereus” and “B. mosaicus”

Line 156.  Italicise “B. mosaicus” (x2)

Line 157.  Italicise “B. cereus”

Line 160.  Italicise “panC”

Line 164.  Italicise “nhe”

Line 165.  Italicise “Sph” and “bpsE”

Line 166.  Italicise “bpsH”

Line 168.  Italicise “cytK2”

Line 169.  Italicise “cesABCD”

Line 170.  Italicise “Vip4Aa1”

Line 171.  Italicise “B. thuringiensis”

Line 173.  Change “The 86%” to “Eighty six percent”

Line 174.  Italicise “vanR-M”, remove “the” before “31%”, italicise “vanZF-Pp” and insert “genes” after “vanZF-Pp”

Line 175.  Italicise “lsa(B)”

Line 176.  Italicise “vanS-Pt2”, “vanR-Pt” and “vanY-Pt2”

Line 177.  Insert “and the” before “tetL” and italicise “tetL”

Line 178.  Change “assessed” to “performed”

Line 179.  Italicise “B. cereus”

Line 181.  Change “to” to “by the”

Line 183.  Italicise “tetL”

Lines 183-184.  Italicise “B. cereus”

Line 184.  Change “result was” to “results were” and “lactama” to “lactam”

Line 185.  Remove “the” before “97%% and “94%”

Line 186.  Remove “respectively”

Line 187.  Italicise “B. cereus”

Line 189.  Change “from 3 ice cream showed each two different” to “from three ice creams that each showed two different” and italicise “B. cereus”

Lines 191-192.  In Table 1 the text is very small and not easy to read, please italicise all bacteria names

Line 192.  Italicise “B. cereus”

Line 193.  Insert “the” before “BTyper”

Lines 194-195.  In Table 1 please italicise all bacteria names

Line 205.  Change “posing” to “poses”

Line 206.  Italicise “B. cereus”

Line 208.  Insert “the” before “B. cereus”

Line 210.  Change “to” to “of”

Line 211.  Remove “the” after “performed”

Line 214.  Insert “of” before “strains”

Line 217.  Insert “the” before “B. cereus”

Line 219.  Change “group” to “groups”

Lines 219-220.  Change “Additional” to “Additionally”

Line 220.  Change “which six are never descripted previous” to “of which six have not been described previously”

Lines 220-221.  Change “The ST more prevalent was” to “The most prevalent ST was”

Line 221.  Change “from the” to “by”

Line 223.  Change “others” to “other”

Line 224.  Remove “respectively” and insert “were” after “STs”

Line 225.  Change “as ST that cause” to “as the ST that causes”

Line 227.  Change “strains” to “and”

Line 229.  Remove “the”

Line 230.  Italicise “nhe” and “sph” and change “with” to “to”

Line 232.  Change “encoding” to “encode” and insert “a” before “high”

Line 240.  The only what?

Line 244.  Change “isolate” to “isolates” and insert “the” before “emetic”

Line 245.  Italicise “ces”

Line 246.  Change “product” to “products” and insert a space before “The”

Line 248.  Change “to five” to “of five”

Line 256.  Change “according to” to “accordance with”

Line 258.  Remove “the”

Line 261.  Change “resistant” to “resistance”

Line 262.  Change “synthesis” to “synthesise”

Line 271.  Change “underline” to “underlined”

Line 272.  Change “genome” to “WGS”?

Line 275.  Change “finding” to “study”

Line 278.  Insert “the” before “B. cereus” and change “reveal” to “revealed”

Line 279.  Change “as” to “to”

Line 282.  Change “the resistant” to “resistance”

Line 283.  Change “and the sensitivity to the other antibiotics” to “and sensitivity to other antibiotics tested”

Line 286.  Insert “the” before “B. cereus”

Lines 293-304.  Suggest using authors’ initials instead of full names to reduce the length of this section

Line 305.  Change “founded” to “funded” and insert “the” before “Ministry”

Lines 311-469.  In the titles of references, change “Bacillus Cereus” to “Bacillus cereus” and italicise “Bacillus cereus”.  Also italicise the names of other bacterial species

Author Response

Dear reviewer,

Thank you very much for taking your time and review our manuscript. Based on your valuable suggestions, a careful revision of the paper has been carried out.
Each of your suggestions has been accepted and the text has been modified.

Reviewer 2 Report

-The paper needs extensive English editing by a native English speaker.

- remove the in Line 17 ( in the 65%) to be in 65%.

- MLVA analysis that was firstly mentioned in the abstract should be written in whole and add abbreviation between brackets, as may be abstract is only read by researchers so they can know the meaning of this abbreviation.

-The same for WGS, AMR and MLST.xample no mentionabout AMR resistance of B. Cereus.

- More identification about the samples should be added, as the quantity of ice cream damples. The brand of ice cream from mrkets is it from one brand or different brans.

- Line 99 you should mention what is the typical colony morphology you selected.

- Kine 106 B. Cereus should be in italic and please correct allover the manuscript.

- introduction need tobe focused on the aim of the study, for example no mention about the AMR of B. Cereus.

- Line 111 what is the meaning by the same food source??

- Line 122 Why you mention the referece , you should change to numer and modify the other reference numbers. The same Line 128.

- Table 3 Line 195, you should mention isolated from ice cream samples.

Author Response

Dear reviewer,

Thank you very much for taking your time and review our manuscript. Based on your valuable suggestions, a careful revision of the paper has been carried out.

 - remove the in Line 17 ( in the 65%) to be in 65%.

The text was modified.

- MLVA analysis that was firstly mentioned in the abstract should be written in whole and add abbreviation between brackets, as may be abstract is only read by researchers so they can know the meaning of this abbreviation.

   The text was modified accordingly.

-The same for WGS, AMR and MLST.xample no mentionabout AMR resistance of B. Cereus.
   The change was applied.

- More identification about the samples should be added, as the quantity of ice cream damples. The brand of ice cream from mrkets is it from one brand or different brans.

  The quantity of ice cream used for the analyis was reported in material and method section. In addition we added more information related to the brand of ice cream samples purchased from supermarket.

- Line 99 you should mention what is the typical colony morphology you selected.
   The suggested change was applied.

- Kine 106 B. Cereus should be in italic and please correct allover the manuscript.
   The suggested change was applied.

- introduction need tobe focused on the aim of the study, for example no mention about the AMR of B. Cereus.
   The text was modified accordingly.

- Line 111 what is the meaning by the same food source??
   We modified the text.

- Line 122 Why you mention the referece , you should change to numer and modify the other reference numbers. The same Line 128.
   we apologize for the error. the references were changed.

- Table 3 Line 195, you should mention isolated from ice cream samples.
   The table was modified.

Reviewer 3 Report

Dear Author,

It is requested to incorporate the following changes

1. In abstract section MLVA, WGS and MLST terms are used. Kindly first write the full name and then abbreviation. 

2. Abstract line No 25-28. kindly rewrite the sentence in order to make it clear for reader.

3. Page 2 line 52 states "on 2019". instead it should be "in 2019"

4. page 2 line 77-78 "To date, the best strategy to characterize members of B. cereus s.l. and the toxin genes, consist in the sequencing of entire genome by next generation sequencing methods [4]. Kindly rewrite the sentence. I could not get it.

5. page 6 line 210 "Messelhausser U. et al. 2010." follow vancouver style of reference citing. 

6 Plagiarism is very high i.e.,36%. Kindly reduce it to 15%.

Regards

Author Response

Dear reviewer,

Thank you very much for taking your time and review our manuscript. Based on your valuable suggestions, a careful revision of the paper has been carried out.

1. In abstract section MLVA, WGS and MLST terms are used. Kindly first write the full name and then abbreviation. 
  The text was modified accordingly.
2. Abstract line No 25-28. kindly rewrite the sentence in order to make it clear for reader.
   The sentence was modified.

3. Page 2 line 52 states "on 2019". instead it should be "in 2019"
   The suggested change was applied.

4. page 2 line 77-78 "To date, the best strategy to characterize members of B. cereus s.l. and the toxin genes, consist in the sequencing of entire genome by next generation sequencing methods [4]. Kindly rewrite the sentence. I could not get it.
    The sentence was modified.
5. page 6 line 210 "Messelhausser U. et al. 2010." follow vancouver style of reference citing. 
     The reference was correctly reported 
6 Plagiarism is very high i.e.,36%. Kindly reduce it to 15%.
    The entire text of the manuscript has been revised and edited.

Round 2

Reviewer 2 Report

Thank the author's for addressing the required corrections